# Diagnostic Performance of a Molecular Assay in Synovial Fluid Targeting Dominant Prosthetic Joint Infection Pathogens

**DOI:** 10.3390/microorganisms12061234

**Published:** 2024-06-19

**Authors:** Jiyoung Lee, Eunyoung Baek, Hyesun Ahn, Heechul Park, Suchan Lee, Sunghyun Kim

**Affiliations:** 1Department of Research & Development, DreamDX Inc., C001, 57, Oryundae-ro, Geumjeong-gu, Busan 46252, Republic of Korea; lab84@dreamdx.net (J.L.); eybaek@dreamdx.net (E.B.); 2Joint & Arthritis Research Center, Himchan Hospital, 120, Sinmok-ro, Yangcheon-gu, Seoul 07999, Republic of Korea; ahs0614@naver.com; 3Department of Clinical Laboratory Science, Hyejeon College, Daehak 1-gil, Hongseong-eup, Hongseong-gun 32244, Republic of Korea; phc2626@hj.ac.kr; 4Department of Clinical Laboratory Science, College of Health Sciences, Catholic University of Pusan, Busan 46252, Republic of Korea; 5Next-Generation Industrial Field-Based Specialist Program for Molecular Diagnostics, Brain Busan 21 Plus Project, Graduate School, Catholic University of Pusan, Busan 46252, Republic of Korea

**Keywords:** prosthetic joint infection, synovial fluid, diagnostic biomarkers, molecular diagnosis

## Abstract

Prosthetic joint infection (PJI) is one of the most serious complications of joint replacement surgery among orthopedic surgeries and occurs in 1 to 2% of primary surgeries. Additionally, the cause of PJIs is mostly bacteria from the *Staphylococcus* species, accounting for more than 98%, while fungi cause PJIs in only 1 to 2% of cases and can be difficult to manage. The current gold-standard microbiological method of culturing synovial fluid is time-consuming and produces false-negative and -positive results. This study aimed to identify a novel, accurate, and convenient molecular diagnostic method. The DreamDX primer–hydrolysis probe set was designed for the pan-bacterial and pan-fungal detection of DNA from pathogens that cause PJIs. The sensitivity and specificity of DreamDX primer–hydrolysis probes were 88.89% (95% CI, 56.50–99.43%) and 97.62% (95% CI, 87.68–99.88%), respectively, compared with the microbiological method of culturing synovial fluid, and receiver operating characteristic (ROC) area under the curve (AUC) was 0.9974 (*** *p* < 0.0001). It could be concluded that the DreamDX primer–hydrolysis probes have outstanding potential as a molecular diagnostic method for identifying the causative agents of PJIs, and that host inflammatory markers are useful as adjuvants in the diagnosis of PJIs.

## 1. Introduction

Prosthetic joint infections (PJIs) are one of the most serious complications of joint replacement surgery [1], and occur in 1 to 2% of patients after primary and 4% of patients after revision surgeries [2]. According to medical claim data from the Korean Health Insurance and Review and Assessment (KHIRA), between 2010 and 2018, the total number of joint replacement surgeries in the Republic of Korea increased from 72,766 to 106,933, PJIs increased from 1786 to 2989 per year, and the incidence of PJIs ranged from 2.3% to 2.8% and steadily increased over time [3,4].

Risk factors that increase the incidence of PJIs include rheumatoid arthritis, obesity, and diabetes. Causes of aseptic failure include screw loosening, fracture of or around the prosthesis, wear, poor implant placement, and poor material [5]. Many patients with knee osteoarthritis (OA) have knee joint disorders. Several types of surgery can be used to treat this disorder, including revision to total knee arthroplasty (TKA) after failed high tibial osteotomy (HTO) or unicompartmental knee arthroplasty (UKA) [6,7].

PJIs are currently diagnosed using the criteria proposed by the Musculoskeletal Infection Society (MSIS) [8]. The MSIS and the Infectious Disease Society developed these criteria to establish a standardized and clear definition of PJIs, improving diagnostic reliability and research collaboration. The major diagnostic criteria include two positive cultures of the same organism, a sinus tract with evidence of communication to the joint, or visualization of the prosthesis. The minor diagnostic criteria include preoperative symptoms, scored per symptom and determined as infected, possibly infected, or uninfected based on elevated serum C-reactive protein (CRP) or D-dimer levels, high serum erythrocyte sedimentation rate (ESR), high synovial white blood cell (WBC) count, synovial leukocyte esterase level (LE), synovial-positive alpha-defensin, high synovial polymorphonuclear cells (PMNs), and high synovial CRP levels. Intraoperative diagnosis is also scored per symptom and determined as infected, inconclusive, or uninfected, according to the score, preoperative score, positive histology, positive purulence, and single positive culture [9,10].

According to the time of onset and symptoms, PJIs are divided into early and delayed infections, which are classified as acute and chronic, respectively. An early infection is an acute PJI with clear signs of local and systemic inflammation. The most prevalent and virulent pathogens, such as *Staphylococcus aureus*, some Gram-negative bacteria, streptococci, and enterococci, cause early infections. Delayed infection causes chronic PJIs, which present with clinical symptoms such as chronic joint pain and premature loosening, and are caused by organisms with low virulence, such as coagulase-negative staphylococci (CoNS) and *Cutibacterium* species [11,12,13]. According to some previous reports [14,15], microorganisms that cause PJIs can be classified into monomicrobial and polymicrobial types, which cover various pathogens, including aerobic bacteria and fungi. The monomicrobial causes of PJIs are primarily *S. epidermidis* and *S. aureus*, followed by *Streptococcus* species and Gram-negative bacilli, enterococci, and CoNS. In another study, the pathogens identified after synovial fluid (SF) culture were CoNS, *S. aureus*, *Streptococcus* spp., Gram-negative rods, *Enterococcus* spp., anaerobes, and other pathogens, including *Candida albicans* and *C. parapsilosis* [14,15]. Additionally, the cause of PJIs is mostly bacteria from the *Staphylococcus* species, accounting for more than 98%, while fungi cause PJIs in only 1 to 2% of cases and can be difficult to manage, and *Candida* species are the most common (85%) [16,17,18].

To diagnose PJIs, microorganisms in SF are cultured and identified; however, this is time-consuming [19]. Therefore, most clinicians use conservative management with empiric antibiotic therapy. Moreover, infections related to microbial causes after total hip arthroplasty (THA) are difficult to diagnose and often require multiple diagnostic methods [20,21]. Currently, PJIs are classified according to the MSIS criteria, and are diagnosed as MSIS-defined culture-positive PJIs when the culture is positive [19]. Despite the availability of various tests, there are limitations in diagnosing PJIs when the culture is negative, and the effectiveness of antibiotics is often limited. Moreover, standardized clinical and evidence-based guidelines for the diagnosis of PJI are lacking; therefore, novel, advanced molecular diagnostic tests would be beneficial [9,22]. To overcome these limitations, various studies have investigated non-culture-based methods. Molecular diagnostic methods, such as quantitative polymerase chain reaction (qPCR), are being established in clinical laboratories and can be combined with existing methods to rapidly, specifically, and sensitively detect causative pathogens of PJIs in SF. However, the above issues have not yet been resolved.

Therefore, the present study aimed to develop a qPCR hydrolysis probe assay targeting bacterial 16S rRNA and fungal 18S rRNA genes to detect and differentiate pan-bacterial and pan-fungal pathogens of PJIs in SF. The result of qPCR hydrolysis probe assay was compared with microbial culture, 16S rRNA and ITS sequence analysis, and host’ acute phase indicators. Further, the study demonstrated that this probe assay is capable of generating rapid, highly sensitive, and specific results for the direct simultaneous detection of bacterial and fungal pathogens.

## 2. Materials and Methods

### 2.1. Patients and Samples

In total, 93 SF clinical samples were obtained from patients (TKA, HTO, or UKA) who had symptomatic osteoarthritis (OA). The samples were collected at Mokdong Hospital (Seoul, Republic of Korea) and Incheon Himchan Hospital (Incheon, Republic of Korea) between January 2022 and May 2023. This study was approved by the Institutional Review Board (IRB) of the Catholic University of Pusan (Approval No. CUPIRB-2022-01-006). Blood samples from patients were retrospectively analyzed using hospital data. SF samples from postoperative patients were aseptically collected using a sterilized syringe and aliquoted into sterilized conical tubes for plate culture, inflammation marker analysis, and DNA quantification. The samples were stored frozen until use. For culture, the SF samples were inoculated on blood agar plates using a sterilized cotton swab and incubated at 37 °C for 18–24 h. Growth covering up to 25% of the plate was considered “few”, up to 75%, “moderate”, and more than 75%, “numerous”.

### 2.2. Genomic DNA Extraction from Reference Strains and SF Samples

Genomic DNA was extracted from reference strains and SF samples using a Gentra Puregene Blood Kit (Qiagen, Hilden, Germany). One milliliter of sample was mixed with five milliliters of cell lysis solution and incubated at 65 °C for 15 min. Then, 30 µL of RNase A solution was added and the samples were incubated at 37 °C for 30 min. After mixing with 2 mL of protein precipitation solution, the samples were cooled on ice for 5 min and centrifuged at 2000× *g* for 10 min to precipitate the proteins. The supernatants were mixed with 6 mL of isopropanol, incubated at room temperature (RT) for 5 min, and centrifuged at 2000× *g* for 10 min to precipitate the DNA. After washing the DNA pellet with 6 mL of 70% ethanol and air-drying for 10 min, 200 µL of DNA hydration solution was added to dissolve the DNA at 65 °C for 1 h. The samples were gently shaken at RT overnight, transferred into 1.5 mL sterilized microcentrifuge tubes (Axygen, Corning, NY, USA), and stored at −20 °C.

### 2.3. Identification of Microorganisms Isolated from SF

Microorganisms in SF were identified using the VITEK 2 system (BioMérieux, La Balme-les-Grottes, France) per the manufacturer’s instructions.

### 2.4. Inflammation Marker Analysis

Serum total protein (TP) and CRP levels, the ESR, and whole-blood and SF WBC counts were determined. Serum TP and CRP levels were measured using a LABOSPECT 006 automatic analyzer (Hitachi High-Tech, Tokyo, Japan). The ESR and whole-blood WBC counts were measured using an XN-1000 hematology analyzer (Sysmex Corporation, Hyogo, Japan).

### 2.5. Bacterial and Fungal Pathogen-Specific Primer and Probe Design

To detect the causative pathogens of PJIs, we designed qPCR primer pairs and hydrolysis probes specific for 16S rRNA and 18S rRNA genes. The 16S rRNA gene target sequence was selected in a conserved region shared by dominant PJI bacterial species, including *S. epidermidis*, *S. aureus*, *Enterococcus faecalis*, *Streptococcus dysgalactiae*, *Streptococcus pyogenes*, *Escherichia coli*, *Acinetobacter baumannii*, *Pseudomonas aeruginosa*, and *Enterobacter cloacae*. The 16S rRNA sequences of *S. epidermidis* (NCBI accession No. NR_036904.1), *S. aureus* (NR_118997.2), *E. faecalis* (MF108410.1), *S. dysgalactiae* (NR_027517.1), *S. pyogenes* (MT535878.1), *E. coli* (NR_024570.1), *A. baumannii* (NR_026206.1), *P. aeruginosa* (NR_026078.1), and *E. cloacae* (NR_102794.2) were aligned to confirm similarity. Similarly, the 18S rRNA gene target sequence was selected in a conserved region shared by dominant PJI fungal species, including *C. glabrata*, *C. albicans*, *C. tropicalis*, and *C. parapsilosis*, and the 18S rRNA gene sequences of *C. glabrata* (JF825465.1), *C. albicans* (LC612900.1), *C. tropicalis* (KX664670.1), and *C. parapsilosis* (LC643454.1) were aligned to confirm similarity (Figure 1 and Appendix A). Primer pairs and hydrolysis probes with slight mismatches were designed using Clustal Omega Tools (https://www.ebi.ac.uk/jdispatcher/msa/clustalo; accessed on 15 June 2024) (EMBL-EBI, Hinxton, UK) and evaluated for secondary structures and dimer-forming potential using the IDT Oligo Analyzer Tools (https://sg.idtdna.com/calc/analyzer; accessed on 15 June 2024) (Integrated DNA Technologies, Coralville, IA, USA). The primer–probe sequence sets exhibiting the fewest secondary structures and lowest dimer-forming potential were selected for use in qPCR and were synthesized at Macrogen Institute (Seoul, Republic of Korea). The forward and reverse primers for pan-bacterial detection amplified 280 bp of the 16S rRNA gene, and the pan-bacterial-specific fluorescent and hydrolysis probes were designed within this region. The primers for pan-fungal detection amplified 161 bp of the 18S rRNA gene, encompassing the pan-fungal-specific fluorescent and hydrolysis probes. Because the causative pathogens of PJIs (9 bacteria and 4 fungi) had similar sequences, bacteria and fungi had to be detected discriminately without overlapping, so the conserved target area was wide, and the amplicon size was relatively large. Also, we had to find common regions in the sequences of the target bacteria and fungi that had no cross-reactivity and had high specificity, respectively. Therefore, hydrolysis probes were designed to increase sensitivity.

### 2.6. qPCR Hydrolysis Probes Assay

The qPCR hydrolysis probe assay (Dream Dx, Busan, Republic of Korea) was optimized to detect the 16S rRNA and 18S rRNA target genes of the dominant PJI-causing bacterial and fungal species. qPCRs were run in 20 μL reaction mixtures containing 10 μL 2 × Thunderbird probe qPCR mixture (Toyobo, Osaka, Japan), 1.0 μL forward primer (10 pmol), 1.0 μL reverse primer (10 pmol), 1.0 μL probe (1 pmol), 2.0 μL template DNA (50–100 ng/µL), and 5.0 μL distilled water. The thermal cycles were as follows: 95 °C for 10 min, 40 cycles of 95 °C for 15 s, 58 °C for 30 s, and 72 °C for 30 s. The efficacy of the DreamDX primer–probe set for the 93 clinical SF samples was confirmed by running qPCRs under the same conditions using reference strains such as ATCC and KCTC (Appendix A) as controls and distilled water as a no-template control (NTC).

In cross-reactivity tests using bacterial and fungal reference strains, results were unreliable at a cycle threshold (Ct) value ≥ 30. Therefore, in this study, the cut-off Ct value was set at 30.

To validate clinical SF samples, the glyceraldehyde-3-phosphate dehydrogenase (*GAPDH*) housekeeping gene was amplified using a primer–hydrolysis probe set [23]. *GAPDH*-negative samples with a Ct value ≥ 30 were omitted from the analysis (Table 1).

### 2.7. Determination of the Limit of Detection (LOD)

The LODs of the respective pan-bacterial and pan-fungal primer–hydrolysis probes set designed in this study were determined using synthetic plasmids for 16S rRNA and 18S rRNA genes (Macrogen Institute) of the nine selected dominant bacterial strains and four selected dominant fungal strains, respectively. The synthetic DNA samples were measured for absorbance using a NanoDrop™ 2000 Spectrophotometer (Thermo Fisher Scientific, Waltham, MA, USA) and gene copies were quantified using a web-based DNA copy number and dilution calculator (https://www.thermofisher.com/kr/ko/home/brands/thermo-scientific/molecular-biology/molecular-biology-learning-center/molecular-biology-resource-library/thermo-scientific-web-tools/dna-copy-number-calculator.html; accessed on 15 June 2024) that computes the number of gene copies based on the measured absorbance.

### 2.8. 16S rRNA and ITS Sequencing Using DreamDX and Universal Primers

The universal primers 27F/1492R and ITS1F/ITS4R (Appendix A) [24,25] were used to compare and confirm the results of microorganism identification of the reference strains and in clinical SF samples. The sequencing was conducted and analyzed at the Macrogen Institute, using Sanger sequencing in an ABI3730xl automated DNA analyzer (Thermo Fisher Scientific).

### 2.9. Analysis of the Specificity of the PJI Molecular Diagnostic Assay

To confirm that the probes detected no non-specific substances in samples, specificity was tested using target and non-target substances. Optimized qPCR hydrolysis probe assay was performed on nine bacterial and four fungal reference strains: ATCC 35989, ATCC 29213, KCTC 3511, KCTC 3098, ATCC 19615, ATCC 35150, KCTC 23254, KCTC 22063, ATCC 2361, KCTC 7653, KCTC 7212, KCTC 7219, and ATCC 10231.

### 2.10. Statistical Analysis

Statistical analyses were conducted using GraphPad Prism version 8 (GraphPad Software, San Diego, CA, USA) to determine differential DNA expression. Receiver operating characteristic (ROC) curves and areas under the ROC curves (AUCs) were used to assess the potential of using SF DNA as a diagnostic sample for PJIs. *p*-values < 0.05 were considered statistically significant.

## 3. Results

### 3.1. Patients and Clinical Samples

Between January 2022 and May 2023, SF samples were collected from 93 patients suspected of PJIs by direct puncture from the knee joints (Table 2). The mean age of the patients was 67.0 ± 13.50 years (range, 20–88 years).

### 3.2. Bacterial Culture Results and Prevalence of Bacterial Pathogens in SF Samples

Among the 93 SF samples, 9 (9.7%) tested positive (CP) in bacterial culture, 84 (90.3%) tested negative (CN) in bacterial culture, and 93 (100.0%) were fungal CN. Among the nine bacterial CP samples, six (66.7%) and three (33.3%) samples comprised Gram-positive (*S*. *aureus*, *S*. *epidermidis*, and *S*. *dysgalactiae*) and Gram-negative bacteria (*E*. *coli*, *S*. *marcescens*, and *E*. *aerogenes*), respectively. Colony counts of the nine bacterial CP samples were “few” in three (33.3%) cases, “moderate” in two (22.2%) cases, and “numerous” in four (44.4%) cases (Appendix A).

### 3.3. LOD of the qPCR Hydrolysis Probes Assay for PJI Diagnosis

To establish a standard curve to evaluate sensitivity, recombinant plasmid DNA concentrations (Appendix A) were established to range from 1 × 10^8^ copies/µL to 1 × 10^1^ copies/µL by 10-fold serial dilution. qPCRs for sensitivity determination were run in duplicate. A standard curve was constructed based on the Ct values for the samples and used to quantify each sample. The standard curves revealed good linearity of amplification, with the following correlation coefficients: *S. epidermidis* (R^2^ = 0.992), *S. aureus* (R^2^ = 0.992), *E. faecalis* (R^2^ = 0.993), *S. dysgalactiae* (R^2^ = 1.000), *S. pyogenes* (R^2^ = 0.995), *E. coli* (R^2^ = 0.998), *A. baumannii* (R^2^ = 0.993), *P. aeruginosa* (R^2^ = 0.999), *E. cloacae* (R^2^ = 0.999), *C. glabrata* (R^2^ = 0.999), *C. albicans* (R^2^ = 0.998), *C. tropicalis* (R^2^ = 0.993), and *C. parapsilosis* (R^2^ = 0.998) (Figure 2). The overall LOD of the assay was 10^2^–10^3^ copies/µL (Table 3).

### 3.4. Specificity of the qPCR Hydrolysis Probe Assay for PJI Diagnosis

The effectiveness and specificity of the DreamDX primer–hydrolysis probe set for pan-bacterial and pan-fungal detection were verified by qPCR hydrolysis probe assay using 13 reference strains. As cross-reactivity between the bacteria and fungi was not observed during target strain detection, all nine bacterial and four fungal reference strains were used as templates for qPCR (Appendix A). The qPCR assay using the pan-bacterial DreamDX primer–hydrolysis probe set amplified the 16S rRNA genes of the nine bacterial reference pathogens and did not amplify the 18S rRNA genes of the fungal reference pathogens, whereas the qPCR assay using the pan-fungal primer–probe set amplified the 18S rRNA genes of the four dominant fungal pathogens, but not the 16S rRNA genes of the bacterial reference strains (Figure 3), indicating high assay specificity for the target bacterial and fungal pathogens. To confirm accuracy, qPCR products for the nine bacterial and four fungal reference strains were sequenced using DreamDX primers for pan-bacterial/fungal detection and the universal primers 27F and 1492R and ITS1F and ITS4R, respectively (Table 4). The results confirmed that the DreamDX primers for pan-bacterial/fungal detection designed in this study and the universal primers amplified all nine bacterial and four fungal reference strains, and the sequences of the products amplified using each primer set completely matched those of the reference strains.

### 3.5. Clinical Utility of the qPCR Hydrolysis Probe Assay for the Detection of Bacterial Pathogens Causing PJIs in Uncultured SF

In molecular biology, housekeeping genes are typically constitutive genes and required in basic cell maintenance function [26,27,28]. To monitor the efficiency of nucleic acid extraction and the potential presence of PCR inhibitors during the analysis, primer and hydrolysis probe sets were designed to detect *GAPDH* gene as an internal control. All 93 SF samples were tested using *GAPDH* primers and hydrolysis probes described elsewhere [23] and validated using *GAPDH* qPCR. In total, 45.16% (42/93) were *GAPDH*-negative and were therefore omitted from analysis, and the 51 (54.84%) *GAPDH*-positive samples were used for analysis.

After validation, based on the culture results and identification using the VITEK 2 system, they were identified as nine (9.68%) bacterial CPs and 42 (45.16%) bacterial CNs. According to the results of DreamDX qPCR for pan-bacterial detection, 10 (10.75%) samples tested bacterial qPCR-positive, and 41 (44.09%) tested qPCR-negative (Figure 4), and DreamDX qPCR for pan-fungal detection yielded negative results for all 51 (54.84%) samples. The sensitivity and specificity of the pan-bacterial DreamDX primers and probe for clinical SF samples were 88.89% (95% CI, 56.50–99.43%) and 97.62% (95% CI, 87.68–99.88%), respectively (Figure 5). ROC analysis showed excellent agreement between the culture and qPCR results (AUC 0.9974; *** *p* < 0.0001), confirming the validity of our assay.

We compared the accuracy of several methods, including the VITEK 2 system and 16S rRNA gene sequencing with the universal primers 27F/1492R and the DreamDX primers for pan-bacterial detection, using nine bacterial CP samples. The VITEK 2 identification results and sequencing results using universal primers and DreamDX primers for pan-bacterial detection were consistent (Table 5). The DreamDX primers for pan-bacterial detection were more effective than the universal primers in identifying bacterial species in clinical SF samples. Although the universal primers targeted a highly conserved 16S rRNA gene region, primer mismatches due to single-nucleotide variants [29,30] may have led to inconsistencies in the sequencing results. The results indicated that the DreamDX primers can be used to effectively detect PJI pathogens.

Finally, comparing the microorganism identification results with the sequencing results using DreamDX and universal primers, the same results were obtained using each primer, making it possible to detect PJI pathogens using DreamDX primers.

### 3.6. Comparison of Host Inflammation Marker Levels between Culture and Molecular Diagnostic Assay-Positive and -Negative Groups

To assess the differences in inflammatory factor levels between bacterial CP and CN samples, we compared TP and CRP levels in the serum, the ESR and WBC count in whole blood, and the WBC count in SF. The AUCs and *p*-values were 0.6042 (*p* = 0.4073) and 0.8000 (** *p* = 0.0035) for serum TP and CRP, respectively; 0.7908 (* *p* = 0.0247) and 0.7136 (* *p* = 0.0118) for ESR and WBC count in whole blood, respectively; and 0.9591 (*** *p* < 0.0001) for WBC count in SF (Figure 6 and Appendix A).

## 4. Discussion

Artificial joint replacement surgery can improve joint function and relieve pain, thus enhancing patients’ quality of life [5]. With the recent trend of population aging, the number of artificial joint replacement surgeries has increased, as has the incidence of PJIs as associated complications.

Infection is currently determined based on the MSIS criteria [31,32]. The current gold standard is SF culture; however, this approach is time-consuming and may produce false-negative and false-positive (indicating the presence of a contaminant) results. Various PCR-based methods for the molecular diagnosis of infectious diseases have been developed and become mainstream [33]. These techniques enable rapid and sensitive detection of PJIs [34], do not suffer from culture contamination, can be useful for detecting uncultured or difficult-to-cultivate bacteria, and can increase the speed, sensitivity, and reliability of clinical diagnosis of causative agents [35]. However, molecular diagnosis remains susceptible to false-positive and false-negative results, and moreover, it is costly [36,37]. Therefore, a novel, accurate, simple, and convenient molecular diagnostic method for PJIs is required. The molecular diagnostic qPCR assay developed in this study required approximately 1.5 h, which is substantially less than the 24–48 h required to obtain a bacterial or fungal CP result from clinical SF samples [38]. The current criteria for diagnosing PJIs are based on both culture results and inflammatory factor levels [2,8,9,10,31,32]. In this framework, the molecular diagnostic assay developed in this study can be used as an auxiliary method.

VITEK 2 analysis of the 93 SF samples collected from two hospitals revealed nine bacterial CP results, namely, two (2.0%) cases of *S. aureus*, two (2.0%) cases of *S. epidermidis*, two (2.0%) cases of *S. dysgalactiae*, one (1.0%) case of *E. coli*, one (1.0%) case of *Serratia marcescens*, and one (1.0%) case of *E. aerogenes* (Appendix A). In line with our findings, *S. aureus*, CoNS, *Streptococcus* spp., *Enterococcus* spp., Gram-negative bacteria, and anaerobes are typical bacterial pathogens that cause PJIs [39].

Around 1–4% of patients who undergo joint replacement surgery are diagnosed with PJIs [2,15,39]. In the present study, nine samples (9.7%) yielded bacterial CP results, and four (4.3%) were diagnosed with PJIs, which agrees with the findings in a previous study [2]. Fungal cultures of the SFs were all negative. Consistent herewith, fungal microorganisms are isolated in less than 1% of PJI cases [40]. Considering these and previous microorganism identification results in SF cultures, a more accurate differentiation of pathogens for PJI diagnosis is clearly needed to allow us to determine whether antibiotics or anti-fungal agents are indicated for treatment [16,17,18]. Metagenomic sequencing allows the analysis of rare microbial species or microbial species that are difficult to culture, and can analyze large amounts of microorganisms simultaneously. However, since information can be obtained from the entire genome, the analysis is costly and time-consuming. On the other hand, 16S rRNA sequencing is a technique that amplifies the target region and analyzes the sequence in parallel in large quantities. It has the advantage of being able to classify microorganisms down to the level of species and analyze them quickly, quantitatively, and at an economical price. In this study, dominant causative pathogens of PJIs (bacteria and fungi) could be detected using the target region of 16S rRNA and 18S rRNA genes, respectively.

In this study, we developed a qPCR diagnostic method to detect bacteria and fungi in SF using 16S rRNA and 18S rRNA genes, respectively, and evaluated its effectiveness. Prior to evaluating the qPCR diagnostic method, we had already established an accurate validation method for clinical SF samples using *GAPDH* as an internal control [23], which was then applied to all SF samples. In another study using *GAPDH* primers as an internal control, qPCR was performed, and nonamplified samples were excluded from case analysis [41]. Since the main causative pathogens (bacteria and fungi) of PJIs have already been identified, the method we developed is more useful for detecting them. Although 16S rRNA, 18S rRNA, and ITS sequence analysis is helpful in identifying bacteria and fungi that are difficult to identify by conventional methods [42,43,44], it has limitations such as the location and size of the amplified base sequence and limitations in discrimination between strains with high genetic identity [45,46]. However, since this is a technique that detects the DNA of the causative agents including bacteria and fungi, it can cause false positives due to DNA from very small amounts of contaminant agents or pathogens killed by already prescribed antibiotics. Therefore, the sensitivity that could be clinically significant was designated as the cut-off value. In addition, when the pathogen multiplied to a certain extent and the Ct was below 30, it could actually be clinically diagnosed as a PJI.

In the current study, the sensitivity and specificity of qPCR for the diagnosis of PJI were significantly higher than those reported in the previous study (73.3%, 95% CI: 66.7–79.2% and 95.5%, 95% CI: 84.5–99.4%, respectively) [47].

In addition, sequencing results using pan-bacterial DreamDX primers and hydrolysis probes were consistently concordant with the VITEK 2 identification data, with 100% sensitivity, specificity, and accuracy. The results of the pan-bacterial DreamDX primer–hydrolysis probe assay developed in this study in terms of CP-dominant PJI species were consistent with those in a previous study [39]. The sequencing and VITEK 2 identification results agreed very well between the pan-bacterial DreamDX and universal primers, with 100% (9/9) and 88.9% (8/9) accuracy, respectively. Potential reasons for inconsistent sequencing results in unmatched samples may include a low number of organisms in the SF, as indicated by the culture results (labeled as “few”) (Table 5), non-uniform distribution of organisms in the SF, PCR inhibitors present in the SF, or inefficient DNA extraction from the samples [48,49].

PJIs induce systemic and local inflammatory and immune responses around the implanted device. TP, CRP, ESR, and WBC count are widely used as host inflammatory markers [50,51]. In this study, serum CRP, whole-blood ESR, WBC count, and SF WBC count showed significant differences between the bacterial CP and bacterial CN groups, while serum TP values did not differ significantly between groups (Figure 5 and Appendix A). Several studies have confirmed the diagnostic and predictive value of serum CRP, whole-blood ESR, WBC count, and SF WBC count as biomarkers [50,52,53].

Therefore, utilizing these host inflammatory markers as supplementary indicators can help to reduce the risk of false positives from skin flora in qPCR tests, enhancing their utility and enabling accurate diagnosis.

qPCR is subject to several limitations, including the potential for contamination. As the Taq polymerase used in qPCR is derived from *E. coli*, residual DNA remaining during the amplification process may be detected as a non-specific amplification product [54]. To overcome this limitation, we used sterile distilled water as a non-template control. Samples were classified as positive or negative based on a Ct cut-off value of 30 in this study. A second limitation specific to the present study was that all SF samples were collected retrospectively. As selection bias is more likely to affect retrospective studies [55], a prospective study using consecutive cases of PJIs should be conducted to offset this limitation [56]. Third, among the bacterial CP samples using SF, one of the three “few” samples exhibited low sensitivity, possibly due to an insufficient amount of template (Table 5) [57,58]. Fourth, although the DreamDX primer–hydrolysis probe set developed for this study is designed for pan-fungal detection and can distinguish between bacteria and fungi, its utility for clinical SF samples remains unconfirmed, as our study did not include fungal CP samples.

In conclusion, this study revealed that the DreamDX primers, designed to be pan-bacterial-specific, were more effective than the universal primers in detecting dominant bacteria causing PJIs. As PJI-causative bacterial pathogens can be detected using DNA extracted from SF, this assay has outstanding potential as a molecular diagnostic method. Additionally, we developed a new molecular diagnostic tool for detecting fungal pathogens in PJIs. However, this assay was tested using only fungal reference strains as a positive control and bacterial reference strains as a negative control as this study did not include clinical fungal-positive SFs; therefore, validation using fungal-positive clinical SF samples is necessary in further studies.

## 5. Patents

The work reported in this manuscript resulted in two patents: Composition for the detection of bacterial infection and detection method using the same (Korean Patent No. 10-2546046, PCT/KR2023/014223) and Composition for the detection of fungal infection and detection method using the same (Korean Patent No. 10-2546048, PCT/KR2023/014246).

## Figures and Tables

**Figure 1 microorganisms-12-01234-f001:**
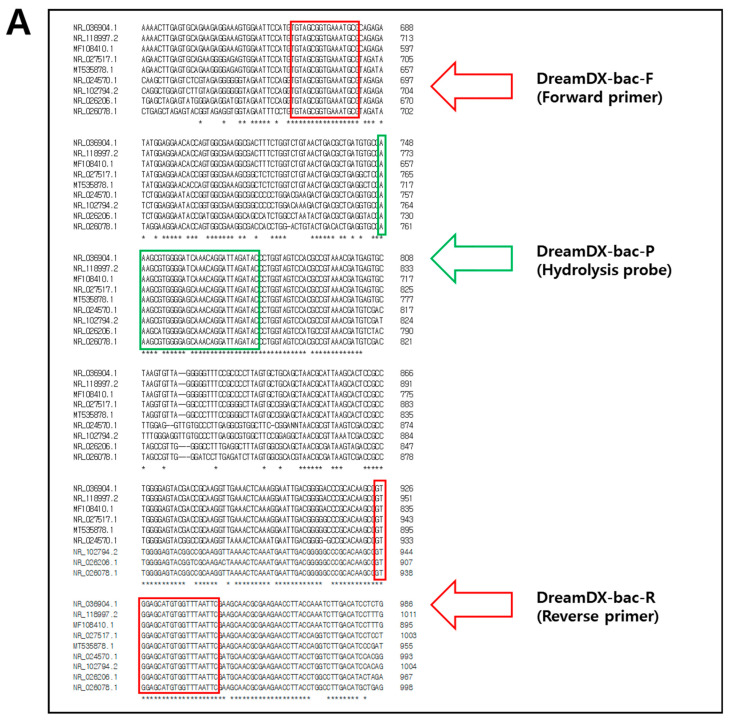
Sequences and positions of (**A**) the nine dominant PJI bacteria-specific PCR primers and probes designed and used in this study for the 16S rRNA gene and (**B**) the four dominant PJI fungi-specific PCR primers and probes designed and used in this study for the 18S rRNA gene. Forward and reverse primer sequences are indicated with red frames, while probe sequences are indicated with green frames. The “*” character means that the residues or nucleotides are indicated identical conserved.

**Figure 2 microorganisms-12-01234-f002:**
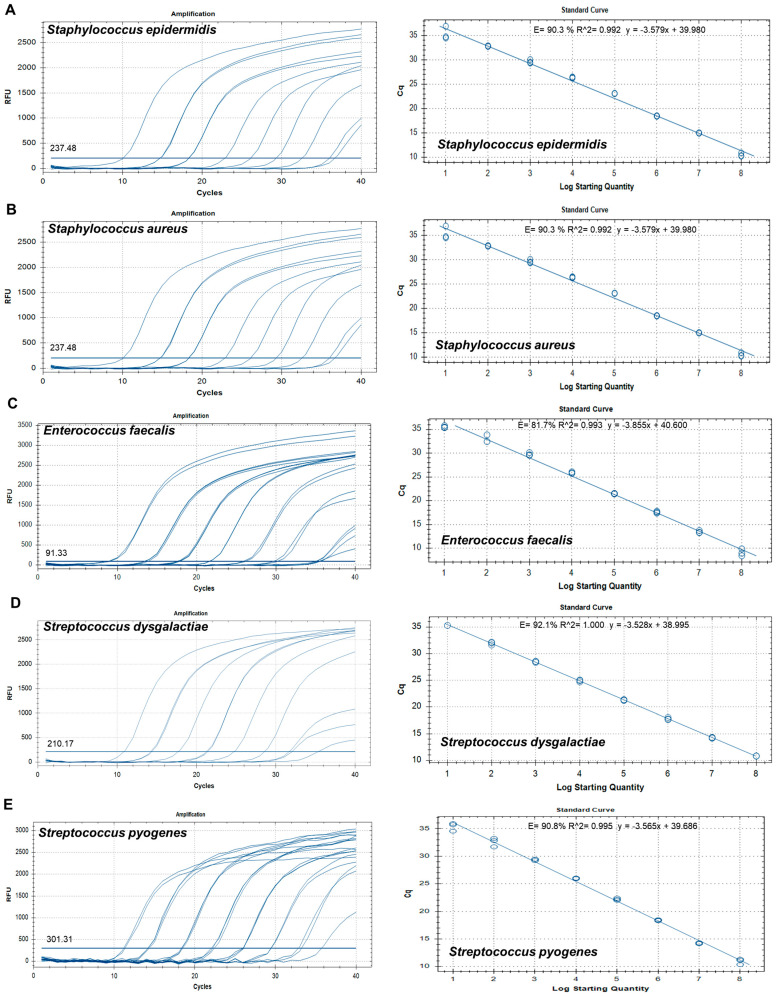
Amplification curves and standard curves of target plasmids with concentrations of 1 × 10^8^ copies/μL to 1 × 10^1^ copies/μL. (**A**) *Staphylococcus epidermidis*, (**B**) *Staphylococcus aureus*, (**C**) *Enterococcus faecalis*, (**D**) *Streptococcus dysgalactiae*, (**E**) *Streptococcus pyogenes*, (**F**) *Escherichia coli*, (**G**) *Acinetobacter baumannii*, (**H**) *Pseudomonas aeruginosa*, (**I**) *Enterobacter cloacae*, (**J**) *Candida glabrata*, (**K**) *Candida albicans*, (**L**) *Candida tropicalis*, and (**M**) *Candida parapsilosis*.

**Figure 3 microorganisms-12-01234-f003:**
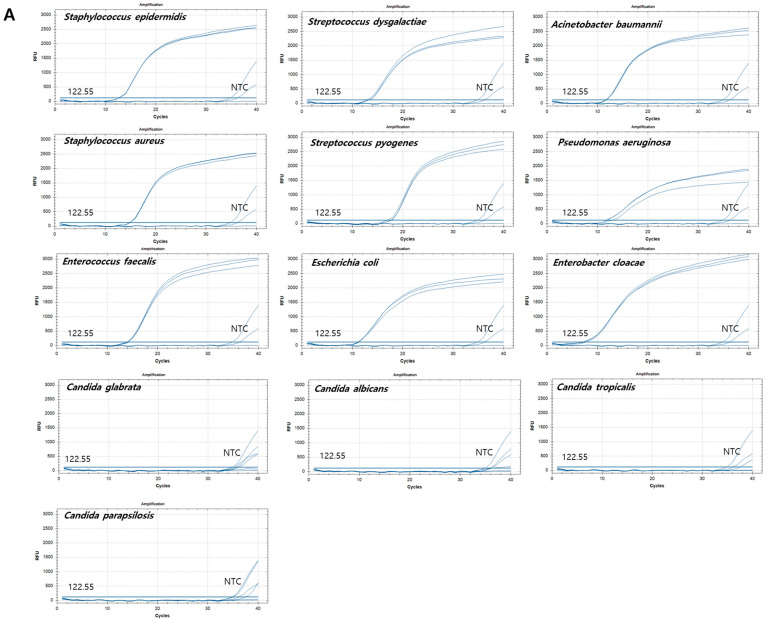
Amplification plots of qPCR for nine bacterial and four fungal reference strains. *X*-axis: cycle numbers; *Y*-axis: relative fluorescence units (RFU). (**A**) Amplification curves generated using the pan-bacterial DreamDX primer–probe set for positive samples (*Staphylococcus epidermidis*, *Staphylococcus aureus*, *Enterococcus faecalis*, *Streptococcus dysgalactiae*, *Streptococcus pyogenes*, *Escherichia coli*, *Acinetobacter baumannii*, *Pseudomonas aeruginosa*, and *Enterobacter cloacae*), negative samples (*Candida parapsilosis*, *Candida tropicalis*, *Candida glabrata*, and *Candida albicans*), and distilled water used as a no-template control (NTC). (**B**) Amplification curves generated using the pan-fungal DreamDX primer–probe set for positive samples (*C. parapsilosis*, *C. tropicalis*, *C. glabrata*, and *C. albicans*), negative samples (*S. epidermidis*, *S. aureus*, *E. faecalis*, *S. dysgalactiae*, *S. pyogenes*, *E. coli*, *A. baumannii*, *P. aeruginosa*, and *E. cloacae*), and distilled water used as an NTC.

**Figure 4 microorganisms-12-01234-f004:**
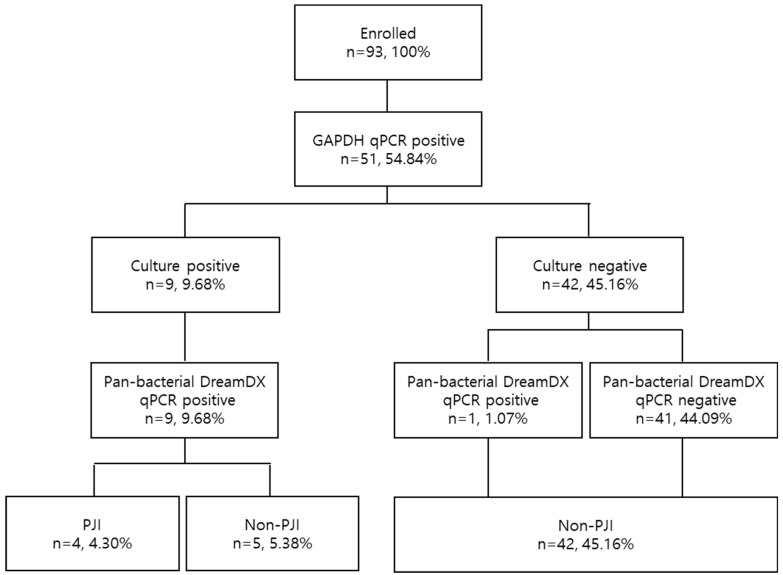
Flow diagram of sample selection based on GAPDH validation, microorganism identification, and pan-bacterial DreamDX qPCR results in PJI and non-PJI groups.

**Figure 5 microorganisms-12-01234-f005:**
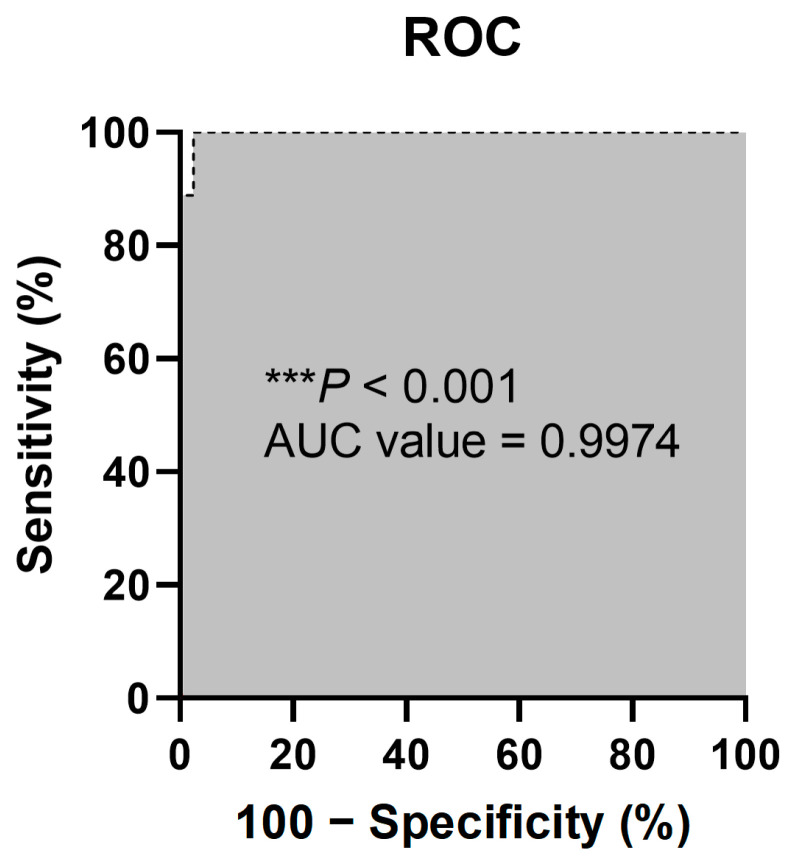
ROC curve of sensitivity (%) versus specificity (%) of the qPCR assay using the DreamDX primer–hydrolysis probe set for pan-bacterial detection of clinical SFs. The “***” character means that the *p* values less than 0.001.

**Figure 6 microorganisms-12-01234-f006:**
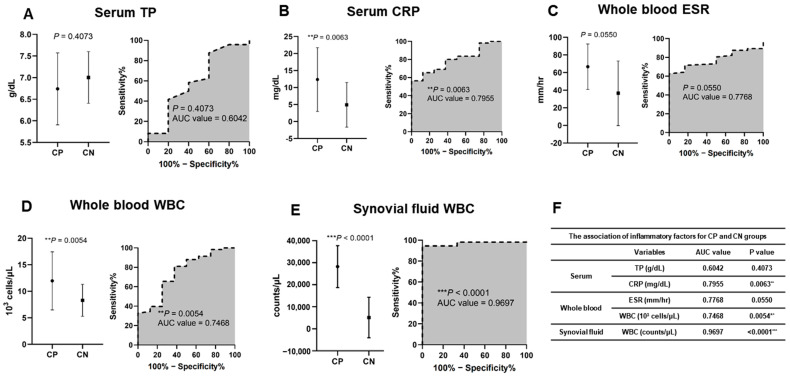
Inflammatory factor levels for CP and CN groups. (**A**) TP in serum, (**B**) CRP in serum, (**C**) erythrocyte sedimentation rate (ESR) in whole blood, (**D**) white blood cells (WBCs) in whole blood, and (**E**) white blood cells (WBCs) in synovial fluid. (**F**) Associations of inflammatory factors.

**Table 1 microorganisms-12-01234-t001:** Validation of clinical SF samples using *GAPDH.*

	*GAPDH*-Positive (Ct < 30)	*GAPDH*-Negative (Ct ≥ 30)	Total
Number of SF samples (*n*, %)	51 (54.84%)	42 (45.16%)	93 (100%)
Ct value (mean ± SD)	21.8 ± 4.28	34.2 ± 2.72	27.4 ± 7.17

Abbreviations: SF, synovial fluid; SD, standard deviation.

**Table 2 microorganisms-12-01234-t002:** Summary statistics of the clinical samples.

		Bacterial CP (*n* = 9, 100%)	Bacterial CN (*n* = 84, 100%)	Total Patients and Fungal CN (*n* = 93, 100%)
Characteristics	Mokdong Himchan Hospital (*n*, %)	8 (88.9%)	46 (54.8%)	54 (58.1%)
	Incheon Himchan Hospital (*n*, %)	1 (11.1%)	38 (45.2%)	39 (41.9%)
	Age (years; mean ± SD)	67.7 ± 19.18	66.9 ± 12.91	67.0 ± 13.50
	Female (*n*, %)	3 (33.3%)	44 (52.4%)	47 (50.5%)
	Male (*n*, %)	6 (66.7%)	40 (47.6%)	46 (49.5%)
OA	Surgically treated group (*n*, %)	8 (88.9%)	40 (47.6%)	48 (51.6%)
	Non-surgically treated group (*n*, %)	1 (11.1%)	44 (52.4%)	45 (48.4%)

Abbreviations: CP, culture-positive; CN, culture-negative; SD, standard deviation; OA, osteoarthritis.

**Table 3 microorganisms-12-01234-t003:** Sensitivity of the qPCR assay.

Organism	Plasmid of Target Strain	Plasmid DNA Concentration	Ct Value	Log Value	LOD Concentration
Bacteria	*Staphylococcus epidermidis*	2.50 fg/μL	29.51	2.93	8.42 × 10^2^ copies/μL
	*Staphylococcus aureus*	2.50 fg/μL	29.51	2.93	8.42 × 10^2^ copies/μL
	*Enterococcus faecalis*	6.00 fg/μL	29.56	2.86	7.31 × 10^2^ copies/μL
	*Streptococcus dysgalactiae*	4.00 fg/μL	28.54	2.96	9.19 × 10^2^ copies/μL
	*Streptococcus pyogenes*	8.00 fg/μL	29.43	2.88	7.53 × 10^2^ copies/μL
	*Escherichia coli*	7.00 fg/μL	28.61	3.01	1.03 × 10^3^ copies/μL
	*Acinetobacter baumannii*	20.0 fg/μL	28.57	3.88	7.55 × 10^3^ copies/μL
	*Pseudomonas aeruginosa*	8.00 fg/μL	28.56	2.96	9.19 × 10^2^ copies/μL
	*Enterobacter cloacae*	7.00 fg/μL	28.14	3.05	1.13 × 10^3^ copies/μL
Fungi	*Candida glabrata*	9.00 fg/μL	28.94	2.97	9.29 × 10^2^ copies/μL
	*Candida albicans*	7.00 fg/μL	28.36	3.00	9.98 × 10^2^ copies/μL
	*Candida tropicalis*	7.00 fg/μL	28.31	2.84	6.91 × 10^2^ copies/μL
	*Candida parapsilosis*	9.00 fg/μL	27.98	8.95	8.85 × 10^2^ copies/μL

**Table 4 microorganisms-12-01234-t004:** Results of sequencing using DreamDX primers for pan-bacterial and pan-fungal detection and universal primers (27F/1492R and ITS1F/ITS4R) for the reference strains.

Reference Strain	Sequencing Primer	Accession No.	Description	Maximum Score	Total Score	Query Cover	E Value	Percent Identified
*Staphylococcus epidermidis* ATCC 35989	DreamDX primer	KY442756.1	*Staphylococcus epidermidis* strain RmS2T 16S ribosomal RNA gene, partial sequence	431	431	41%	1 × 10^−115^	99.16%
	Universal primer	LC648288.1	*Staphylococcus epidermidis* IRQBAS126 gene for 16S rRNA, partial sequence	2058	2058	98%	0.0	99.38%
*Staphylococcus aureus* ATCC 29213	DreamDX primer	OQ626118.1	*Staphylococcus aureus* strain 344 16S ribosomal RNA gene, partial sequence	429	429	50%	3.00 × 10^−115^	98.37%
	Universal primer	AP028993.1	*Staphylococcus aureus* TUM19959 DNA, complete genome	113	682	6%	9 × 10^−20^	94.52%
*Enterococcus faecalis* KCTC 3511	DreamDX primer	MK791584.1	*Enterococcus faecalis* strain CE_1_5 16S ribosomal RNA gene, partial sequence	425	425	59%	3 × 10^−114^	97.60%
	Universal primer	HG799973.1	*Enterococcus faecalis* partial 16S rRNA gene, isolate OCAT41	1984	1984	96%	0.0	96.67%
*Streptococcus dysgalactiae* subsp. *equisimilis* KCTC 3098	DreamDX primer	CP053074.1	*Streptococcus dysgalactiae* subsp. *equisimilis* strain TPCH-A88 chromosome, complete genome	401	2411	28%	1 × 10^−106^	96.37%
	Universal primer	CP053074.1	*Streptococcus dysgalactiae* subsp. *equisimilis* strain TPCH-A88 chromosome, complete genome	1825	10,948	99%	0.0	99.70%
*Streptococcus pyogenes* ATCC 19615	DreamDX primer	LS483391.1	*Streptococcus pyogenes* strain NCTC8320 genome assembly, chromosome: 1	442	2654	10%	2 × 10^−118^	99.59%
	Universal primer	HG316453.2	*Streptococcus pyogenes* H293, complete genome	1567	9374	97%	0.0	96.35%
*Escherichia coli* O157: H7 ATCC 35150	DreamDX primer	CP123592.1	*Escherichia coli* strain DJH_137 urine chromosome, complete genome	427	2953	57%	1 × 10^−114^	97.98%
	Universal primer	MT279578.1	*Escherichia coli* strain E.Sa.12 16S ribosomal RNA gene, partial sequence	2050	2050	95%	0.0	97.36%
*Acinetobacter baumannii* KCTC 23254	DreamDX primer	CP096717.1	*Acinetobacter baumannii* strain 5741 chromosome, complete genome	433	2571	25%	5 × 10^−116^	98.77%
	Universal primer	LN611357.1	*Acinetobacter baumannii* partial 16S rRNA gene, isolate 4031	2043	2043	97%	0.0	97.20%
*Pseudomonas aeruginosa* KCTC 22063	DreamDX primer	MK348744.1	*Pseudomonas aeruginosa* strain EC8A23 16S ribosomal RNA gene, partial sequence	424	424	6%	1 × 10^−112^	97.23%
	Universal primer	HM439417.1	*Pseudomonas aeruginosa* strain PCP33 16S ribosomal RNA gene, partial sequence	226	226	89%	9 × 10^−54^	72.19%
*Enterobacter cloacae* ATCC 2361	DreamDX primer	CP040827.1	*Enterobacter cloacae* strain NH77 chromosome, complete genome	407	3236	53%	1 × 10^−108^	97.90%
	Universal primer	MN244518.1	*Enterobacter cloacae* strain CP21 16S ribosomal RNA gene, partial sequence	2174	2174	95%	0.0	98.16%
*Candida glabrata* KCTC 7219	DreamDX primer	OP150922.1	*Candida glabrata* isolate D2RB internal transcribed spacer 1, partial sequence	207	207	23%	2 × 10^−48^	96.83%
	Universal primer	ON398731.1	*Candida glabrata* isolate UDSMC8_ITS-1 internal transcribed spacer 1, partial sequence	1537	1537	67%	0.0	100%
*Candida albicans* ATCC 10231	DreamDX primer	KP675137.1	*Candida albicans* strain H373A internal transcribed spacer 1, partial sequence	211	211	38%	9 × 10^−50^	99.15%
	Universal primer	OK267909.1	*Candida albicans* clone 13–20 internal transcribed spacer 1, partial sequence	928	928	85%	0.0	99.41%
*Candida tropicalis* KCTC 7212	DreamDX primer	OP143792.1	*Candida tropicalis* isolate D4SDB internal transcribed spacer 1, partial sequence	211	334	62%	8 × 10^−50^	99.15%
	Universal primer	AB467294.1	*Candida tropicalis* gene for ITS1, strain: TL0301	817	817	98%	0.0	96.22%
*Candida parapsilosis* KCTC 7653	DreamDX primer	MW703799.1	*Candida parapsilosis* isolate IPFUNGSarsCov2-25 small subunit ribosomal RNA gene, partial sequence	215	215	64%	4 × 10^−51^	97.64%
	Universal primer	MT153735.1	*Candida parapsilosis* isolate 4Y137 internal transcribed spacer 1, partial sequence	887	887	71%	0.0	99.59%

**Table 5 microorganisms-12-01234-t005:** Ct values and sequencing results for the clinical SF samples.

No.	Organism	Quantity of Colonies	Ct Value (Mean ± SD)	Sequencing Primer	Accession No.	Description	Maximum Score	Total Score	Query Cover	E Value	Percent Identification
1	*Staphylococcus aureus* (Bacteria)	numerous	26.1 ± 0.03	DreamDX primer	OQ626099.1	*Staphylococcus aureus* strain 266 16S ribosomal RNA gene, partial sequence	409	409	11%	2 × 10^−108^	98.29%
				Universal primer	MH447063.1	*Staphylococcus aureus* strain B0020-03R 16S ribosomal RNA gene, partial sequence	1306	1306	88%	0.0	90.72%
2	*Staphylococcus aureus* (Bacteria)	numerous	24.3 ± 0.3	DreamDX primer	OQ626118.1	*Staphylococcus aureus* strain 344 16S ribosomal RNA gene, partial sequence	412	412	37%	4 × 10^−110^	97.15%
				Universal primer	AP023034.1	*Staphylococcus aureus* TPS3156 DNA, complete genome	320	1895	24%	5 × 10^−82^	86.05%
3	*Streptococcus dysgalactiae* (Bacteria)	numerous	17.0 ± 0.11	DreamDX primer	CP053074.1	*Streptococcus dysgalactiae* subsp. *equisimilis* strain TPCH-A88 chromosome, complete genome	427	2566	24%	2 × 10^−114^	98.36%
				Universal primer	OP263260.1	*Streptococcus dysgalactiae* subsp. *equisimilis* strain RERA134 16S ribosomal RNA gene, partial sequence	1903	1903	92%	0.0	95.29%
4	*Escherichia coli* (Bacteria)	moderate	20.5 ± 1.32	DreamDX primer	CP088766.1	*Escherichia coli* strain B16EC0875 plasmid pB16EC0875-2, complete sequence	440	440	25%	3 × 10^−118^	99.59%
				Universal primer	KP276714.1	*Escherichia coli* strain UT1 16S ribosomal RNA gene, partial sequence	1122	1122	51%	0.0	92.69%
5	*Staphylococcus epidermidis* (Bacteria)	few	29.2 ± 0.88	DreamDX primer	OR740675.1	*Staphylococcus epidermidis* strain CM334 16S ribosomal RNA gene, partial sequence	436	436	10%	1 × 10^−116^	99.17%
				Universal primer	FJ835669.1	Uncultured bacterium clone YO00379D02 16S ribosomal RNA gene, partial sequence	479	479	23%	8 × 10^−130^	99.25%
6	*Staphylococcus epidermidis* (Bacteria)	few	25.9 ± 0.99	DreamDX primer	OR740675.1	*Staphylococcus epidermidis* strain CM334 16S ribosomal RNA gene, partial sequence	431	431	94%	4 × 10^−116^	98.77%
				Universal primer	CP043804.1	*Staphylococcus epidermidis* strain SESURV_p4_1553 chromosome	1192	7120	62%	0.0	95.81%
7	*Serratia marcescens* (Bacteria)	moderate	20.3 ± 0.00	DreamDX primer	MN493799.1	*Serratia marcescens* strain MKBn5e 16S ribosomal RNA gene, partial sequence	438	626	50%	1 × 10^−117^	98.79%
				Universal primer	OM757735.1	*Serratia marcescens* strain SEM 16S ribosomal RNA gene, partial sequence	1334	1334	75%	0.0	90.56%
8	*Streptococcus dysgalactiae* (Bacteria)	numerous	17.3 ± 0.10	DreamDX primer	CP053074.1	*Streptococcus dysgalactiae* subsp. *equisimilis* strain TPCH-A88 chromosome, complete genome	433	2599	29%	4 × 10^−116^	99.17%
				Universal primer	EF121441.1	*Streptococcus dysgalactiae* strain 155719-2 16S ribosomal RNA gene, partial sequence	1842	1842	85%	0.0	96.06%
9	*Klebsiella aerogenes* (Bacteria)	few	24.3 ± 2.09	DreamDX primer	KU308267.1	*Klebsiella aerogenes* strain B8 16S ribosomal RNA gene, partial sequence	438	438	78%	3 × 10^−118^	99.18%
				Universal primer	KT998836.1	*Klebsiella aerogenes* strain TIL_WAK_137 16S ribosomal RNA gene, partial sequence	1247	1247	68%	0.0	96.24%

## Data Availability

The original contributions presented in the study are included in the article/supplementary material; further inquiries can be directed to the corresponding authors.

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
