# Peer review of "Diagnostic Performance of a Molecular Assay in Synovial Fluid Targeting Dominant Prosthetic Joint Infection Pathogens"

_microorganisms, 2024, doi:10.3390/microorganisms12061234_

Round 1

Reviewer 1 Report

Comments and Suggestions for Authors

This study aimed to develop a novel, accurate, and convenient molecular diagnostic method for PJIs.  It is a very interest and meaningful work. It is suggested to be accepted after mini revision.

(1) When comparing this method with metagenomic sequencing, what are the advantages and disadvantages? It is suggested that the authors describe this in the discussion.

(2) The quality of Figure 1 and Figure 3 should be improved.

(3) Line 408, E. coli should be italic, please check the whole manuscript!

Author Response

Thank you very much for taking the time to review this manuscript. Please find the detailed responses below and the corresponding revisions in the re-submitted files. Please see the attachment.

Reviewer 2 Report

Comments and Suggestions for Authors

The work has indisputable relevance. Diagnostic optimization and technological improvement are encouraged in microbiology.

Despite their relevance, some points need to be corrected or improved:

1- When citing causes of aseptic failure, the authors mistakenly list “microbial infection” (Lines 45 and 46);

2- In the paragraph that goes from line 65 to 82, the authors need to write the specific terms of microorganisms in italics;

3- It is important to replace the commercial term “TaqMan” by the nature of the reagent, in this case, hydrolysis probes (lines 26, 28, 31, 98, 140, 141, 154, 160, 162, 170, 185, 186, 191, 237, 257, 258, 262, 289, 294, 295, 313, 385, 387, 388 and 417);

4- It is interesting that there is an explanation for the relatively large size of the 16S rRNA amplicon (line 159);

5- The figures need to be enlarged, with the exception of figure 5;

6- The cut-off Ct value was very low (lines 182 and 183). This meant that almost half of the samples were not suitable for analysis. It would be interesting to explain a little more about the criteria for this decision and make clear the consequences in relation to the need for synovial fluid new collection;

7- In line 186, the “>” sign must be replaced by “≥”;

8- In lines 233 and 234, the terms specific to microorganisms must be in italics;

9- As there is no variation in the information in the “Primer set” column of table 3, this column could be removed from the table;

10- In line 257, the term “QPCR” would be better presented as “qPCR”;

11- The term “PCR” would be better presented as “qPCR” in lines 378, 380, 407 and 408;

12- I suggest reducing the font in tables 4 and 5. When table 5 was divided, the separation made it difficult to understand the sectioned information;

13- If it is a reference, the “5” must be arranged as “[5]”, on line 345;

14- If it is a reference, the “31” must be arranged as “[31]”, on line 389;

15- The term “E. coli” must be in italics (line 408);

16- Lines 444 to 449 appear not to have been filled out correctly.

Author Response

(The authors gave the same response as above.)

Reviewer 3 Report

Comments and Suggestions for Authors

The paper by Jiyoung Lee et al. addresses an exciting subject. I express my sincere gratitude for the opportunity to review your manuscript. The effort of the author is appreciated. The manuscript is well written, and the review protocols and data are reported scientifically and presented clearly. Congratulations on your results. 

As a reviewer, the title of the manuscript "Diagnostic Performance of a Molecular Assay in Synovial Fluid Targeting Dominant Prosthetic Joint Infection Pathogens" led me to anticipate a study encompassing both knee and hip prosthetic joint infections (PJIs). However, upon reviewing the Materials and Methods section, it appears that the cases studied are confined to total knee arthroplasty (TKA) and unicompartmental knee arthroplasty (UKA), with the inclusion of high tibial osteotomy (HTO) cases, which might more appropriately be classified as fracture-related infections (FRIs) rather than PJIs. To align the study's focus with its title, I recommend either revising the title to more accurately reflect the content—potentially excluding reference to PJIs—or amending the methodology. This adjustment will enhance clarity and ensure the manuscript accurately represents the scope of the study.

The names of bacteria sahould be formatted according to binomial nomenclature rules, which dictate that the genus name is capitalized and the species name is presented in lowercase, with both parts italicized to denote scientific terms. Upon subsequent mentions in a document, the genus name can be abbreviated to its initial, which is followed by a period, maintaining the italicization to ensure clarity of reference. Details concerning specific strains or subspecies, which follow the species name, are not italicized, differentiating them from the primary binomial name. These formatting rules are crucial for maintaining accuracy and uniformity in the communication of scientific information. Please assess the indroduction section of the manuscript.

Author Response

(The authors gave the same response as above.)

Round 2

Reviewer 3 Report

Comments and Suggestions for Authors

Dear Authors,

The revised version of your manuscript has significantly improved in quality. In my opinion, it is now ready to proceed to the next stage of the publication process.